# Tuning Localized Surface Plasmon Resonance of Nanoporous Gold with a Silica Shell for Surface Enhanced Raman Scattering

**DOI:** 10.3390/nano9020251

**Published:** 2019-02-12

**Authors:** Wei Li, Chao Ma, Ling Zhang, Bin Chen, Luyang Chen, Heping Zeng

**Affiliations:** 1Shanghai Key Laboratory of Modern Optical System, Engineering Research Center of Optical Instrument and System (Ministry of Education), School of Optical-Electrical and Computer Engineering, University of Shanghai for Science and Technology, Shanghai 200093, China; liwei_usst_329@163.com (W.L.); machao_usst@163.com (C.M.); hpzeng@phy.ecnu.edu.cn (H.Z.); 2State Key Laboratory of Metal Matrix Composites, School of Materials Science and Engineering, Shanghai Jiao Tong University, Shanghai 200030, China; steelboy@sjtu.edu.cn; 3Key Laboratory for Ultrafine Materials of Ministry of Education, School of Materials Science and Engineering, East China University of Science and Technology, Shanghai 200237, China; chenly@ecust.edu.cn

**Keywords:** localized surface plasmon resonance (LSPR), nanoporous gold (NPG), surface-enhanced Raman scattering (SERS)

## Abstract

We report the tuning of localized surface plasmon resonance (LSPR) of nanoporous gold (NPG) by silica coating, which also affects the surface enhanced Raman scattering (SERS) of NPG. In this study, controllable silica shell is assembled on the NPG surface, and a fully silica thin layer causes more than 50 nm red-shift of LSPR band due to dielectric medium dependence. Additionally, ~1 nm silica coated NPG film shows excellent SERS enhancement, which is due to electromagnetic coupling between ligaments and local surface plasmon field enhancement within pores, and theoretical analysis indicates that silica coating further improves the coupling effect, which demonstrates the electromagnetic origin of the tuning of SERS effect.

## 1. Introduction

Numerous nanoporous metals (NPM) that were obtained via dealloying exhibit a bicontinuous network and have been demonstrated to possess superior optical properties [1,2,3,4]. Among these, nanoporous gold (NPG) has stood apart for use in analytical and sensing applications, such as surface enhanced Raman scattering (SERS) [5,6,7], surface enhanced fluorescence (SEF) [8,9,10], and bio-chemo-sensing [1,11,12]. This can be attributed to its many desirable properties, including excellent chemical stability, tunable pore size, high effective surface area, and strong electromagnetic fields (EM) appearing in the vicinity of metallic ligaments. Recently, the SERS ability of free-standing NPG films have been systematically investigated, not only due to the high specific surface area and the interconnected hollow channels [5,12,13,14,15], but also because of the controllable localized surface plasmon resonance (LSPR) related with ligament and nanopore sizes. According to the research, NPG with smaller ligaments and nanopore size exhibits better SERS enhancement, and the relatively strong enhancement results from the hotspot due to the large curvatures of the nano-sized ligaments and the EM coupling between neighboring ligaments and intensified LSPR [6,7,8,9,16,17,18]. However, for the bare NPG with small ligaments, the LSPR band is around 545 nm [19]. In order to further tune the LSPR of NPG, a core-shell structure has been constructed with the LSPR band shifts red to ~620 nm [20,21]. Since the LSPR and coupling effects are the key to the NPG-based SERS substrates, in this study, we fabricated silica coated NPG (SiO_2_@NPG) [22,23,24,25], and systematically investigated the influence of silica shell on LSPR and the related SERS effect. The LSPR band seriously varied with silica shell thickness, and the SERS ability of SiO_2_@NPG substrate reached a maximum value with around 1–2 nm-thick silica coating, which is two times stronger than that of the bare NPG. The use of a chemically inert shell coating around the NPG not only can protect the SERS-active nanostructure from contact with whatever is being probed [26], but also extend the range of resonance excitation wavelength due to the tuning of NPG LSPR around few tens of nanometers. The enlarged SERS activity with longer wavelength laser excitation expanded the possible application of NPG-based substrate in the biochemical sensing field.

## 2. Materials and Methods

NPG films were prepared from Au_35_Ag_65_ (at.%) alloy films with a thickness of ~100 nm [27,28]. Polymer supported alloy films were immerged in nitric acid for 20 min at room temperature. The bare NPG films were washed several times by ultrapure water (18.25 MΩ·cm) to remove excess nitric acid, and then put in a drying cabinet for further experimentation.

Atomic layer deposition (ALD) is a widely used method for forming uniform and ultrathin solid films, including metal oxides and semiconductor materials with sub-nanometer thickness [2,21]; however, heating processes are always needed for vapor-phase ALD [21]. In order to avoid a coarsening effect of NPG during heating; here, we adopt another wet chemical method to coat NPG with thin silica. A surfactant template assistant method was applied to prepare a silica shell, and Figure 1 shows the fabrication process. NPG films were immerged in 0.1 M cetyltrimethylammonium bromide (CTAB) aqueous solution for 1 h to assemble CTAB on the surface of NPG. Then, CTAB modified NPG films were stabilized in the plating solution, which was mixed by 50 mL 6 mM CTAB, 25 mL ethanol (99.7%) and 30 μL NH_3_·H_2_O at 35 °C with steadily stirring. Five minutes later, 80 μL tetraethoxysilane (TEOS) was added dropwise in the container under vigorous stirring to ensure homogeneity of the solution, silica was formed by hydrolysis of TEOS [22]. CTAB modified NPG films were kept in solution for 2 h, 5 h, 10 h, 20 h and 30 h, respectively, and silica shells with different thickness were generated on the NPG surface. The thickness of silica was increased with time but not with a linear dependence. Finally, the SiO_2_@NPG films were washed by ethanol and ultrapure water alternately, and then dried in the air. For comparison, the silica shell was coated on gold nanoparticles (AuNPs) decorated on polymer films with the same method described in Figure 1. The gold AuNPs were prepared by laser ablation, which has been reported in our previous work [29]. 

The SERS measurements were carried out on a homemade Raman spectroscope with a 50 mm spectrograph (Andor) and CCD (Andor). A droplet of crystal violet (CV) aqueous solution was dropped on as-prepared substrate and covered with coverslip for SERS dection. Both 532 nm and 633 nm laser excitation were used for the Raman measurements, and the size of laser beam is ~5 μm in diameter with a power of 0.6 mW on the sample surface and 10 s exposure time.

## 3. Results and Discussion

The microstructure of NPG and SiO_2_@NPG films were observed by scanning electron microscope (SEM, ThermoFisher Scientific, Massachusetts, MA, USA) and transmission electron microscope (TEM, JEM-2100F, JEOL, Tokyo, Japan). Figure 2a shows the SEM image of the bare NPG, and the average nanopore size of NPG is ~28 nm in diameter. From the TEM images (Figure 2b,c) of SiO_2_@NPG, it can be seen that silica fully covers the surface of NPG ligaments, and the thickness of the silica layer increases with the plating time. With 10 h of plating, the thickness of silica is ~1–2 nm (Figure 2b), while, with 20 h of plating, the thickness of silica is ~5–10 nm (Figure 2c). More evidence can be found from the Energy Dispersive Spectrometer (EDS), where the mass percentage of Si increases with the reaction time, while Au decreases gradually (Figure 2d). Thus, the thickness of the silica layer can be adjusted by controlling the reaction time, and it can also be noted that the silica film thickness is dependent on the shape of the pores, and different thickness is attained in spherical and elliptical pores. 

The back-scattering spectra were accomplished by a UV-Vis spectrophotometer (HORIBA Dual-FL-UV-800, Kyoto, Japan). Extinction spectra of NPG and SiO_2_@NPG films are shown in Figure 3a. There are two characteristic peaks in each spectrum [30], and the shorter wavelength peak located at about 490 nm does not change with silica coating because this peak originates from the resonance of gold films [19,20,31,32], whereas the longer wavelength peak, arising from LSPR, represents red-shift from ~550 nm to ~610 nm with the increase of silica shell (Figure 3b), due to the larger refractive index of silica compared to air [20,21]. Additionally, the LSPR band dramatically changed after 5 h plating indicating that NPG is becoming perfectly covered. Compared to silica coated Au NPs (see Appendix A), the LSPR tuning of NPG via silica coating is more obvious than that of nanoparticles, where the extinction spectra exhibit only a slight red-shift after silica coating [25,26]. 

In order to investigate the silica effect on an NPG based SERS substrate, we measured the SERS activity of SiO_2_@NPG films with both 532 nm and 633 nm laser excitation. The SERS spectra were evaluated using CV as probe molecules and a droplet of 2.5 µL CV aqueous solution with the concentration of 10^−5^ M was dripped on the substrate, and then covered with a coverslip for measurement. Figure 4a,c,e show the SERS spectra of CV molecules on the substrates, and several characteristic peaks of CV range from 1000 to 1800 cm^−1^ can be observed. The normalized SERS intensity of Raman peaks of CV at 1178, 1369 and 1617 cm^−1^ with different silica coating times is analyzed and listed in Figure 4b,d,e. With 633 nm laser excitation, silica coating first leads to a slight decrease of SERS intensity; however, further coating with silica causes an obvious increase of SERS intensity due to the tuning of the LSPR wavelength to 610 nm, which is consonant with the excitation wavelength [26]. The SERS intensity gradually reached a peak value with the 10 h plated NPG, at which the SERS enhancement is twice that of bare NPG. After this, the SERS intensities decrease gradually due to the further increased inter-distance between the CV molecules and NPG surface. With 532 nm laser excitation (Figure 4c,d), the signal intensities decrease with the silica coating, indicating that the LSPR plays a key role for the SERS of an NPG substrate. 

For comparison, SERS ability of silica coated Au nanoparticles (SiO_2_@AuNPs) was also evaluated using CV as probe molecules, and the detection process was similar to the NPG films. Figure 4e presents the SERS spectra of CV (10^−4^ M) molecules on AuNP decorated polymer films. The SERS intensities (Figure 4f) obtained from SiO_2_@AuNPs are weaker than the ones from SiO_2_@NPG. Those characteristic peak intensities dramatically decreased with the silica coating, unlike SiO_2_@NPG where a maximum point exhibits with ~2 nm silica shell by 633 nm laser excitation. The different performance of NPG is mainly attributed to the LSPR variation that related to the unique bicontinuous nanostructure and coupling effect of adjacent ligaments. 

The electromagnetic enhancement origins of the SERS effect with SiO_2_@NPG as substrate were further investigated by finite-difference time-domain (FDTD) simulation, and the electromagnetic distribution of both bare NPG and silica-coated NPG were analyzed. A simplified 2D ring structure with identical ligament (D) and pore (d) sizes (Figure 5a) is introduced to qualitatively simulate the electromagnetic filed of NPG [33], and the characteristic length of ring is set as 28 nm in circular diameter, which is approximately equal to the ligament size of NPG that was used in the experiment. Moreover, a uniform silica thin layer is covered on the NPG surface to simulate SiO_2_@NPG, and the boundary condition is a perfectly matched layer (PML) and the fineness of the grid is 1 nm × 1 nm × 1 nm. To obtain the electromagnetic field in the vicinity of ring interface, an *x*–*y* plane monitor is placed at the position of the central section, and the simulation time is set at 1000 fs to ensure complete decay of the field. Under the plane wave with wavelength of 633 nm and horizontal direction polarized propagating along the direction normal to the top surface of the nanostructure, the local electromagnetic distributions are shown in Figure 5b–g. Strong electromagnetic fields that related to the polarization direction of input source [34] appear in the vicinity of gold ligaments, and the inner intensities are stronger than the outside ones, which is due to the coupling between ligaments. Moreover, with the increasing of silica thickness, the intensity of electromagnetic fields exhibits a maximum value at the point of 2 nm silica (see Figure 5h), which matches well with the experimental results. Generally speaking, the coating ultrathin silica shell on the NPG surface can tune the LSPR wavelength as well as promote the intensity of SERS, and both the experimental and theoretical results confirm that the high sensitivity achieved with SiO_2_@NPG is attributed to the long-range effect of the enhanced field that was generated by strengthening of the coupling effect and index induced focus [35].

## 4. Conclusions

In summary, the NPG surface has been coated with ultrathin silica films by a chemical plating method with the thickness adjusted by the reaction time. The resonant LSPR band presents a remarkable red-shift with silica coating. Compared with core-shell nanoparticles, NPG films exhibit an excellent SERS effect after coating with silica, which is due to electromagnetic coupling between ligaments and local surface plasmon field enhancement within pores. This study has an important implication in developing the SiO_2_@NPG films as an SERS sensing candidate.

## Figures and Tables

**Figure 1 nanomaterials-09-00251-f001:**
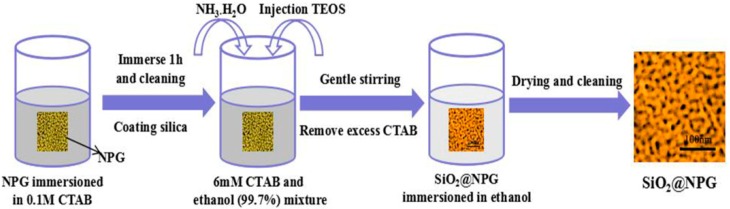
Schematic diagram of the SiO_2_ layer fabrication.

**Figure 2 nanomaterials-09-00251-f002:**
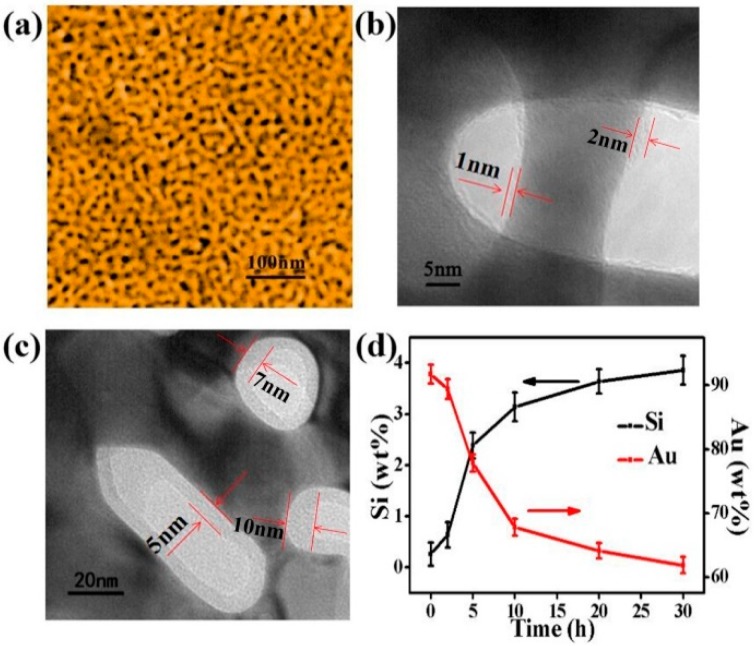
(**a**) Scanning electron microscope (SEM) image of nanoporous gold film with the nanopore and ligament sizes of ~28 nm; (**b**) Transmission electron microscope (TEM) micrograph of SiO_2_@NPG film with 1–2 nm SiO_2_ coating (10 h plating); (**c**) TEM micrograph of SiO_2_@NPG film with 5–10 nm SiO_2_ coating (20 h plating); (**d**) the ratio of the relative content of gold and silicon with different SiO_2_ coating time, detected by Energy Dispersive Spectrometer (EDS).

**Figure 3 nanomaterials-09-00251-f003:**
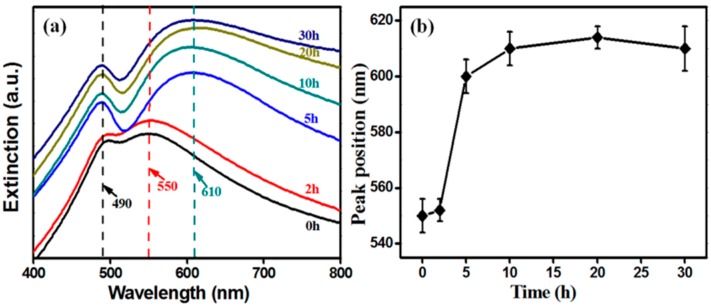
(**a**) extinction spectra and (**b**) the localized surface plasmon resonance (LSPR) wavelength (higher peak) position of SiO_2_@NPG films with different silica coating times (0 h represents bare NPG).

**Figure 4 nanomaterials-09-00251-f004:**
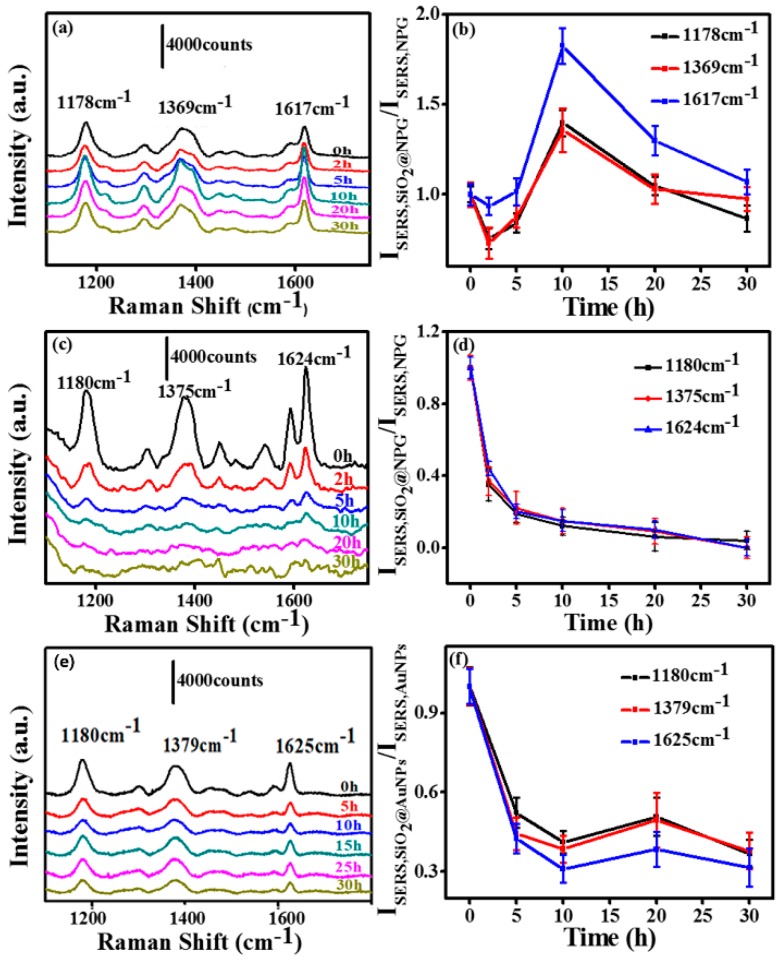
(**a**) and (**c**) surface enhanced Raman scattering (SERS) spectra of crystal violet (CV) molecules on SiO_2_@NPG films with silica coating time 0 h, 2 h, 5 h, 10 h, 20 h, 30 h (laser excitation: (**a**) 633 nm and (**c**) 532 nm); (**b**) the normalized SERS intensity of Raman bands of CV at 1178, 1369 and 1617 cm^−1^ on SiO_2_@NPG films shown in (**a**); (**d**) the normalized SERS intensity of Raman bands of CV at 1178, 1369 and 1617 cm^−1^ on SiO_2_@NPG films shown in (**c**); (**e**) SERS spectra of CV molecules on AuNPs and SiO_2_@AuNPs (at indicated coating time) with 633 nm laser excitation; (**f**) the normalized SERS intensity of Raman bands of CV at 1180, 1379 and 1625 cm^−1^ on SiO_2_@AuNPs films.

**Figure 5 nanomaterials-09-00251-f005:**
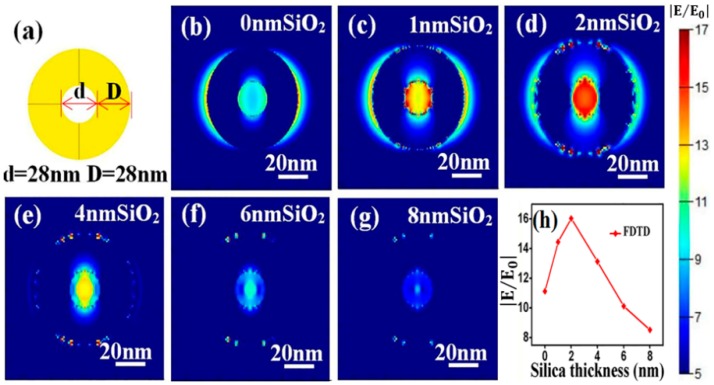
(**a**) three-dimensional ring structure for FDTD simulation. (**b**–**g**) local electromagnetic distributions on the top surface of SiO_2_ coated NPG; (**h**) variation of electric field intensity with silica thickness.

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
