# Peer review of "Tuning Localized Surface Plasmon Resonance of Nanoporous Gold with a Silica Shell for Surface Enhanced Raman Scattering"

_nanomaterials, 2019, doi:10.3390/nano9020251_

Reviewer 1 Report

Please find attached my report.

Author Response

Thank you for the kindly suggestion, we rewrote the manuscript and added some additionally experimental descriptions and analysis according to the reviewer's comments.

Reviewer 2 Report

The manuscript by Li et al. is reports an incremental study on the fabrication of nanoporous gold films coated with thin layers of silicon dioxide by sol-gel method. In my opinion, the manuscript needs considerable improvement to be accepted for publication. Therefore, I cannot recommend publication of this manuscript in its current form. The manuscript needs mayor revision to be considered in Nanomaterials. Here are some recommendations for the Authors consideration:

- English editing is needed. The manuscript is readable but, though I understand the Authors are not native speakers, there are serious grammatical and style errors throughout the manuscript that need to be fixed before the manuscript can be considered for publication

- The manuscript has no detailed experimental section. There are many experimental details that should be included, from fabrication and characterisation to simulations.

- The relationship silica coating thickness–deposition time should be included in the manuscript in the form of graph or table.

- In my opinion, it would be more appropriate to say “plasmonic band” than “peak”.

- This sentence makes no sense: “Additionally, the resonant peak dramatically changed with 5 hours plating indicates NPG was perfectly covered, and the thickness of silica shell seldom affects the SPR of NPG since the reflective index is constant”.

- Typo in concentration of CV (10^-5M).

- The Authors should provide a detailed explanation of what “light bending” means in the context of metallic nanostructures coated with a thin layer of semiconductor oxide.

Author Response

Thank you for the kindly suggestion, we rewrote the manuscript and added some additionally experimental descriptions and analysis according to the referee’s comments. 

Reviewer 3 Report

The authors report on the effect of silica coating on the optical extinction and SERS oriented applications of nanoporous gold (NPG) silica coatings.

Although the topic is timely, coherent with the journal’s scope and the experimental evidence potentially very interesting, the manuscript, in its present form, should not be published

The manuscript is really difficult to read. This is mainly due to:

a)Poor English

b)Poor sentence structuring

On the scientific side the underling physical explanation is, in general, more suggestive than based on strict arguments (the only exception being represented by the FDTD simulations). Just as an example let me quote one of the authors explanations: 

“Silica coating caused large optical loss of AuNPs due to the positive curvature edges and the peak intensities decrease continuously, whereas a maxima point exhibits at 10h plating NPG that due to unique bicontinuous nanostructure and coupling effect of adjacent ligaments. Therefore, Raman signal still have obvious enhancement after coated with more than 5 nm silica shell.”

Were the authors do make an effort to rationalize some of their findings with more stringent arguments (FDTD simulations reported in Figure 5), the matching between the experimental findings and the simulations is claimed with no reference to the specific experimental data (which graph, which data point?). Furthermore, no explanation on how the simulation are performed is presented.

The authors should make an effort to better contextualize their work. For instance, a brief survey on alternative techniques and their pro and cons would be desirable for a broader audience. This is particularly important since, as reported by the authors, the SERS enhancement, where present, amounts sonly to a factor of 2, see Figure 4. The investigated sample could still be advantageous, but the advantages have to be clearly stated.

The temperature increase an its spatial profile, an issue of the utmost importance when dealing with metallic nano-objects under focused laser excitation (such as in SERS experiments), is not mentioned. I suggest the authors to address this point together with the sample thermal stability.

It would really help the reader if the authors reported the layer thickness vs the reaction time.

The overall impression is that of a potentially interesting work, with nice measurements but where no effort was made to cast the text in a readable form nor to interpret the experimental findings in a clear, stringent way. For the above-mentioned reasons I cannot recommend the manuscript for publication.

Author Response

(The authors gave the same response as above.)

Reviewer 4 Report

In this manuscript, the nanoporous gold (NPG) film covered with silica was applied to SERS substrate. The SERS signal from the SiO2-coated NPG film was stronger than that from the bare NPG film because of the higher refractive index of SiO2. It is very interesting topic. But, there are some problems as follows. 

In the introduction, the SiO2-coated noble metal nanoparticles should be mentioned, because they correspond the NPG film via Babinet's principle. 

The SiO2-coated Au nanoparticles was compared with the SiO2-coated NPG. For the former, the excitation wavelength and the extinction spectra (like Fig.3a) should be shown. The trend of Fig.4f may be possibly explained by the LSPR peaks in the extinction spectra rather than the positive curvature edges. Moreover, the average diameter of the nanoparticle should be written. If it is the same as the nanopore size, the Au nanoparticles can correctly correspond to and be compared with the NPG film. 

In the text, the authors must use subscript or superscript for the wavenumbers or the number of the atoms in the moleucular formulas, respectively. 

In the line 80 and 82: The words as “lower” and “higher” should be replaced with “shorter” and “longer”, respectively. 

Author Response

Thank you for the kindly suggestion, we rewrote the manuscript and added some additionally experimental descriptions and analysis according to the referee’s comments.

Round  2

Reviewer 2 Report

The Authors have made efforts in addressing my comments and suggestions. Though I still consider that the presented results in this manuscript and fundamentally incremental, it could be of some interest for the Audience of Nanomaterials. Publication can be granted.

Author Response

The Authors have made efforts in addressing my comments and suggestions. Though I still consider that the presented results in this manuscript and fundamentally incremental, it could be of some interest for the Audience of Nanomaterials. Publication can be granted.

Response:We thank the reviewer for his/her positive and insightful comments and finding our work of interest and suitable for publication in Nanomaterials. We also appreciate the essential summary by the reviewer of our work, which is helpful to further elucidate the background of our work and to improve our presentation.

Reviewer 3 Report

In blue you’ll find my current review: comment/requests.

In black you’ll find my past criticisms and requests that are not sill addressed.

In red the authors responses to past requests

The authors partially succeeded in redacting the text in a more comprehensible way, nevertheless english still needs to be improved. It is not a matter of style but rather a matter of correct english writing. 

I urge the authors to take further action in this sense.

Original referee request: Figure 5), the matching between the experimental findings and the simulations is claimed with no reference to the specific experimental data (which graph, which data point?). 

This point is still missing an answer.

Original referee request: Furthermore, no explanation on how the simulation are performed is presented

The authors just added the fact that they implmented PML. This fact alone does not provide an explanation and discussion of the model implementation.

Original referee request:The authors should make an effort to better contextualize their work. For instance, a brief survey on alternative techniques and their pro and cons would be desirable for a broader audience. This is particularly important since, as reported by the authors, the SERS enhancement, where present, amount sonly to a factor of 2, see Figure 4. The investigated sample could still be advantageous, but the advantages have to be clearly stated.

The authors still do not state in clear the advantages of their sample over exisiting ones. 

Original referee request. The temperature increase (spatial profile), an issue of the utmost importance when dealing with metallic nano-objects under focused laser excitation (such as in SERS experiments), is not mentioned. I suggest the authors to address this point together with the sample thermal stability.

Response 5: Thank you for the comment. The thermal effect could be ignored because the exposure time is 10 sencond and the laser power on the sample surface is about 0.6mW. We added the experimental setup in the revised manuscript.

The authors still do not address the point. 

With respect to the temperature what matters is not the average laser power but the absobed power per unit volume. 

The absobed power per unit volume requires some extensive work to be extracted, newetheless the authors should at least provide 

the impinging power/unit surface area (they give the average power, what is the spot size of the laser beam once focused on the sample?) 

and, possibly: (b) an estimate of the absorbed power based on an estimate of the absorption coefficient of their material at the laser wavelength they use.

Refer for instance to the paper Danesi et al.: Phys. Chem. Chem. Phys., 2018, 20, 15307—15315 to grasp what the problemi is in a similar expreiment on somewhat similar situation.

An other possibility is characterising the sample before and after the exposure to ensure nothing changed. This is somewhat a more difficult task.

Minor corrections:

Figure 4 needs to be enlarged. It’s barely readable.

Summary

The authors did ameliorate the manuscript, nevertheless they did not address quite a few points I made in my previous report.

This points have to be throughly addressed before the manuscript can be published.

Author Response

We thank the reviewer for his/her positive and insightful comments and finding our work of interest and suitable for publication in Nanomaterials. 

Reviewer 4 Report

In the response to my comments, the authors show the absorption spectra of AuNPs with the different laser ablation time. However, it does not correspond with Fig.3a. In this case, the spectra of silica-coated AuNPs with the different coating time should be shown in Supporting Information. Then, the trend of Fig.4f should be discussed like Fig.4b/d which related with Fig.3a. Fig.4f may be possibly explained by the LSPR peaks in the extinction spectra rather than the positive curvature edges. 

Author Response

Thank you for the comments. 
